# Relationship Between Urban Year-Round Green Exercise and Perceived Health, Well-Being, and Reasons for Engagement

**DOI:** 10.3390/ijerph22101562

**Published:** 2025-10-14

**Authors:** Konrad Reuß, Christopher Huth

**Affiliations:** Chair of Sports and Health Management, Institute for Sports Science, University of the Bundeswehr Munich, 85579 Neubiberg, Germany; christopher.huth@unibw.de

**Keywords:** green exercise, meteorological conditions, health benefits, wellbeing, reasons for engaging

## Abstract

Urban year-round green exercise (YRGE)—defined as moderate to vigorous physical activity performed regularly in natural urban settings throughout all seasons and weather conditions—has the potential to promote health, well-being, and social connectedness. This study investigates the relationship between YRGE and individuals’ perceived health status, psychological well-being, and reasons for engagement. A cross-sectional online survey was conducted with 408 adult participants engaged in urban green exercise. The findings indicate that physical activity in adverse meteorological conditions, such as rain, cold, and wind, is positively associated with perceived current health, health over the past 12 months, and well-being. Social connectedness is particularly influenced by environmental factors like sun exposure and heat. The study also reveals key motivational factors for YRGE participation, including improving health and fitness, disconnecting from everyday life, enjoying nature, and experiencing tranquility, with significant variation depending on age and individual nature connectedness. These results suggest that YRGE serves as an accessible and inclusive public health intervention with consistent benefits across socio-demographic groups. Urban planning and health promotion initiatives should prioritize the maintenance and accessibility of urban green spaces and offer guided YRGE programs to encourage sustainable participation across the population.

## 1. Introduction

Natural environments are reported to have physiological health-promoting effects, such as a reduction in blood pressure or cortisol levels, and well-being-enhancing effects, including improved self-confidence and overall well-being [1,2,3]. These effects can be further enhanced when a stay in nature is combined with physical activity (PA), also known as green exercise (GE) [4]. GE has been defined as PA practiced in natural settings [5,6] and is said to have greater effects on health and well-being for all, regardless of socio-economic factors, compared with PA in other environments [7,8,9]. According to theories such as attention restoration theory (ART) [10] and the stress reduction theory (SRT) [11,12], individuals who regularly engage with nature tend to exhibit higher levels of psychological and physiological benefits, improved cognitive capacities, or tend to be more satisfied with life and overall well-being [2,13]. However, in a world that is rapidly urbanizing, access to nature and natural settings, such as urban green spaces (UGS), is becoming increasingly complex [14]. Besides structural influences, such as proximity and accessibility to UGS [15,16,17], GE is strongly influenced by adverse weather conditions, including precipitation or cold temperatures, leading to a substantial decrease [18,19,20]. This decline means that the individual health benefits are lost. The spatial proximity of UGR alone is not decisive for the amount of GE [21]. The theory of planned behaviour [22] suggests that attitude towards the behaviour, subjective norms, and perceived behavioral control can influence behaviour. GE performed throughout the year can counteract this decline. GE performed throughout the year, under all weather conditions, with moderate to vigorous intensity, will be referred to as year-round green exercise (YRGE) [23]. Therefore, this study aims to assess the relationship between YRGE and people’s perceived health, well-being, social connectedness, as well as people’s reasons for engagement and potential group differences, such as in socio-demographic factors or their connectedness with nature. This will be done by the following questions, which will guide this research:

Is there a correlation between YRGE and the perceived health, well-being, and social connectedness, and are there differences between different groups?

Furthermore, what are people’s reasons for engaging in YRGE, and are there differences between different groups?

## 2. Materials and Methods

Since GE is strongly reduced during meteorological conditions such as rain, snowfall, or cold, some of the questions were explicitly aimed at meteorological conditions that are generally perceived as bad weather [18]. The participants received a brief explanation of the different weather conditions. For example, cold was defined as being below 0 degrees Celsius, heat was defined as above 30 degrees Celsius, and strong wind was defined as more than 40 km/h. The questionnaire was adapted from a previous study [24] and revised to record the perceived health benefits and the reasons people have for engaging in YRGE. The questionnaire was tested beforehand for clarity of wording and logical structure and revised accordingly by the authors. The internal consistency of the individual questions was checked using Cronbach’s alpha. Where possible and appropriate, a 5-point Likert scale ranging from (1) disagree to (5) agree was used to answer the questions. These scales have been widely used since they best reflect the participants’ perspectives [25,26]. The questionnaire was available online from 15 January 2024 to 15 March 2024, and was available in the German language. Data was collected online via the SoSci Survey online platform version 3.5.07. The participant’s recruitment was based on the snowball principle. Therefore, the online survey platform link was distributed through social media and sent to sports federations, requesting they further distribute it to sports clubs and their members. No eligibility criteria were applied other than being older than 18 years and engaging in GE in UGS. No incentives for participation were provided.

The questionnaire contained the following. The Section 1 included demographics such as age, gender, educational level, and income. As well as general questions on urban YRGE and participants’ connectedness to nature [16]. The following section asked participants to explain their reasons for engaging in urban YRGE. Therefore, providing statements such as “I am physically active in urban green spaces because I want to improve my health” or “I want to experience a contrast to the built environment” [24]. In the subsequent section, participants were asked to report their perceived current health status, well-being, and social integration.

### Data Analysis

772 questionnaires were completed, of which 408 could ultimately be used for the evaluation. The remaining questionnaires were excluded due to missing data. The data set was initially analyzed descriptively, and a series of Spearman rank correlations and Pearson’s correlations were conducted. ANOVA was used to calculate differences between groups, with Bonferroni post-hoc tests and effect sizes classified according to Cohen’s guidelines. The data was analyzed using IBM SPSS Statistics Version 29.0.2.

## 3. Results

### 3.1. Participant’s Characteristics

Among the participants, 53.9% (n = 220) are female, and 46.1% (n = 188) are male. On average, participants are between 41 and 50 years old, with a net monthly income ranging from €2001 to €3000. A total of 58.6% (n = 239) indicated they hold a university degree (bachelor’s, master’s, or doctorate. Additionally, 59.3% (n = 242) described their work as predominantly sedentary. 96.3% (n = 393) of the participants reside in Germany, and a majority of 86.3% (n = 352) had access to UGS within 15 min. On average, participants engaged in GE in UGS 2–3 times per week (µ = 2.88, SD = 1.37) with moderate to vigorous intensity. The participants’ nature-connectedness score had an average of 3.6 (SD = 0.8). Participants mostly engaged in endurance sports, such as running or Nordic walking (51.5%, n = 220). Detailed descriptions of the participants can be found elsewhere [16].

### 3.2. Perceived Health, Well-Being, and Social Benefits

Participants perceived their current health status as rather good (µ = 3.25, SD = 0.53). The same was found for their health over the last 12 months (µ = 3.15, SD = 0.55), their well-being (µ = 3.12, SD = 0.61) and their social connectedness (µ = 3.29, SD = 0.62). Being active during various meteorological conditions, especially during adverse weather, was associated with perceived current health, health over the past twelve months, and well-being. Social connectedness tends to be influenced primarily by factors such as precipitation, heat, and sun exposure. Effect sizes were found to be small to medium. Table 1 shows the correlations. No differences were found regarding the type of activity participants engaged in, nor for socio-demographic factors.

No differences were found in health statuses, well-being, and social connectedness across socio-demographic factors.

### 3.3. Reasons for Engaging in Urban Year-Round Green Exercise

For 55.7% (n = 227) of participants, having fun is a (somewhat) important motivation. Experiencing nature is a (somewhat) important factor for 63% (n = 257), with significant differences regarding individuals’ nature connectedness (F(20,705) = 11.330, *p* < 0.001). Pushing and discovering physical limits is a (somewhat) important reason for 49.2% (n = 201), with significant differences between age groups (F(7,400) = 5.59, *p* < 0.001), with a medium effect size (n^2^ = 0.089). Weight reduction and improving physical appearance are a (somewhat) important motivation for 32.8% (n = 134), with a significant difference between age groups (F(7,400) = 5.588; *p* < 0.001). This aspect is less important for older participants (µ = 2.13; SD = 1.02) than for younger ones (µ = 4.0; SD = 1.05). Maintaining fitness and improving health is a (somewhat) important reason for 82.9% (n = 338). Everyday safety is a (somewhat) important factor for 27.2% (n = 111), with a significant difference observed between age groups (F(7,400) = 5.138; *p* < 0.001). This aspect is more important for older individuals (µ = 3.75, SD = 1.06) than for younger ones (µ = 2.11, SD = 1.37). Maintaining social contacts is a (somewhat) important motivation for 46.1% (n = 188), with significant age-related differences (F(7,400) = 3.738; *p* < 0.001). Social contact is more important for older participants (µ = 4.38, SD = 0.95) than for younger ones (µ = 2.88, SD = 1.43). Experiencing tranquility is a (somewhat) important motivation for 62.8% (n = 256) with significant differences regarding individuals’ nature connectedness (F(22,7141) = 11.974, *p* < 0.001), with a large effect size (n^2^ = 0.180). Disconnecting from everyday life is a (somewhat) important reason for 82.1% (n = 335), with significant differences regarding individuals’ nature connectedness (F(15,932) = 9.075, *p* < 0.001), with a medium to large effect (n^2^ = 0.137). Experiencing a contrast to the built environment is a (somewhat) important motivation for 49.7% (n = 203), with significant differences regarding individuals’ nature connectedness (F(16,715) = 9.457, *p* < 0.001), with a large effect size (n^2^ = 0.142). No differences were found for the remaining socio-demographic factors or the type of sports participants engaged in. Table 2 highlights these findings.

No differences were found for the remaining socio-demographic factors or the type of sports participants engaged in.

## 4. Discussion

This research aimed to examine the relationship between urban YRGE and perceived health and well-being. The goal was to expand the understanding of how people engage in YRGE. Building on previous work on YRGE, this research aimed to examine aspects that are not bound to infrastructure or natural elements, but rather are inherent in the individual’s perception. This study identified a correlation between urban YRGE and both health and well-being. People who remain active during adverse weather conditions seem to benefit from this activity. Specifically, during weather events such as rain, cold, and wind, GE is associated with higher perceived current health, better health over the past year, and increased perceived well-being. Social connectedness appears to be influenced by heat, to a lesser extent by other weather factors. Therefore, this finding suggests that urban YRGE can be considered a reliable health intervention, which is supported by theories such as ART as well as SRT [10,12]. Since no differences were observed across socio-demographic groups, urban YRGE can be viewed as a helpful and accessible health option for everyone [8,23]. Given the significant drop in GE during adverse weather conditions, especially in winter [20], the study’s results suggest that sports organizations, federations, and local governments should encourage urban YRGE to maximize its health benefits. To promote urban YRGE effectively as a health strategy, understanding why people participate can provide valuable insights. The main reasons include disconnecting from daily routines and maintaining physical fitness and health. Other motivators include having fun, experiencing nature, and finding tranquility. The first two align well with the expected health benefits of GE [2] and are reflected in participants’ perceptions of their health. Since perceived benefits can influence behavior [22] these findings can foster an understanding of why people engage in GE during adverse weather conditions. Since no differences were found between socio-demographic factors, these findings suggest that YRGE is a form of GE that appears to benefit everyone. A similar finding from this study is that the type of sports practiced does not have an impact. However, these findings need to be interpreted with caution since most participants engaged in a similar type of exercise. Overall, enjoying nature appears to be a strong motivator for engaging in urban YRGE, with no notable differences across socio-demographic groups. This suggests that emphasizing nature experiences should be a priority for sports organizations, cities, or municipalities when promoting health programs. This can be achieved by expanding UGS, making them easily accessible, or offering guided courses and programs. Providing programs or upgrading sports facilities can also benefit younger individuals, who often participate in urban YRGE to discover their physical limits, lose weight, or improve their appearance. Additional differences were seen among older adults, for whom maintaining social connections and safety in daily life are especially important. Both of these factors could benefit from guided courses and programs offered by cities and sports federations. These findings highlight the importance of guided courses and programs for urban YRGE, particularly considering the apparent scarcity of such offerings [23].

## 5. Conclusions

According to the results of this study, a relationship appears to exist between urban YRGE and perceived health and well-being, which seems to hold regardless of socio-demographic factors. Therefore, YRGE can be seen as a form of health intervention which is beneficial for all. YRGE tends to be influenced by nature and through personal experience. Maintaining and improving UGS should therefore be a priority for cities and governments, with additional guided courses and programs for urban YRGE.

## 6. Limitations

This study has some limitations. Among these limitations is the use of online surveys as a convenience sampling method, which might limit the generalizability of the findings. Furthermore, participants tend to have a relatively high level of education and a strong connection to nature. The latter might explain participants’ high value of nature in their reasons for engaging in urban YRGE. Another limitation is that this study relies on participants’ self-reported health and well-being, which may be subject to bias. Future studies should consider these limitations by focusing on groups with a lower socio-economic status and people who are less engaged in YRGE. Furthermore, data gathering took place in the spring, which may lead to a distortion in participants’ recall of being active in the summer heat.

## Figures and Tables

**Table 1 ijerph-22-01562-t001:** Spearman rank correlation between being active during different meteorological conditions and perceived health, psychological well-being, and social connectedness.

Meteorological Condition	Current Health Status	Health Status over the Last 12 Months	Psychological Well-Being	Social Connectedness
Precipitation	0.176 **	0.125 *	0.099 *	0.109 *
Cold (<0 degrees Celsius)	0.202 **	0.196 **	0.195 **	0.080
Wind (>40 km/h)	0.102 *	0.113 *	0.132 **	0.044
Heat (>30 degrees Celsius)	0.131 **	0.107 *	0.081	0.138 **
Sun radiation (UV-index > 3)	0.096	0.085	0.084	0.102 *

Note: * *p* < 0.05; ** *p* < 0.01.

**Table 2 ijerph-22-01562-t002:** Importance of reasons for engagement with differences between groups and effect sizes.

Reason for Engagement	Important for Participants	Group Differences	Effect Size
Having fun	55.7% (n = 227)	Non	-
Experiencing nature	63% (n = 257)	Nature connectedness	small
Pushing physical limits	49.2% (n = 201)	Age	medium
Weight reduction/physical appearance	32.8% (n = 134)	Age	small
Maintaining fitness/improving health	82.9% (n = 338)	Non	-
Everyday Safety	27.2% (n = 111)	Age	small
Maintaining social contact	46.1% (n = 188)	Age	small
Experiencing tranquility	62.8% (n = 256)	Nature connectedness	large
Disconnecting from everyday life	82.1% (n = 335)	Nature connectedness	medium
Experiencing contrast from build environment	49.7% (n = 203)	Nature connectedness	large

## Data Availability

The data supporting the findings of this study are available from the corresponding author upon reasonable request.

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
