# Peer review of "Relationship Between Urban Year-Round Green Exercise and Perceived Health, Well-Being, and Reasons for Engagement"

_ijerph, 2025, doi:10.3390/ijerph22101562_

Round 1

Reviewer 1 Report

Comments and Suggestions for Authors

The introduction section is concise but repetitive, written in a less formal style, as opposed to academic language. At the end of this section, the research questions address the differences between various groups. Except for age, what are the other parameters that define different groups? It will be of interest to mention the criteria that differentiated the groups.

In the Methods section, the Cohen effect size is mentioned. Except for a brief mention in the text, it is not used in data analysis. Perhaps the authors could include the effect size in the table or use it to nuance the somewhat overused term in the Results section.

In the Results section, there is a confusing sentence: The average age range of participants is 41 to 50 years. The average or mean is a value, while the range is an interval. 

Gathering data in a cold season psychologically encourages participants to recall recent impressions and less the experience at temperatures of more than 30 degrees C. In my opinion, it could be a bias. 

The discussion section is briefly covered, with the most cited article of your own. Perhaps it can be explained by the novelty of your approach or the scarcity of literature on green exercise...

Including the Abbreviation section at the end of the article will be helpful for readers.

Author Response

Dear Reviewer 

We thank you for your constructive and helpful comments on the manuscript. We have tried to implement them to your satisfaction.

The changes that we conducted are highlighted in yellow within the manuscript.

Comment 1: The introduction section is concise but repetitive, written in a less formal style, as opposed to academic language. At the end of this section, the research questions address the differences between various groups. Except for age, what are the other parameters that define different groups? It will be of interest to mention the criteria that differentiated the groups.

Response 1:We added information about the groups. This can now be found in Lines 52 & 53

Comment 2: In the Methods section, the Cohen effect size is mentioned. Except for a brief mention in the text, it is not used in data analysis. Perhaps the authors could include the effect size in the table or use it to nuance the somewhat overused term in the Results section.

Response 2: We added the effect size to some of the relevant results. This can now be found in lines 140, 143 & 146.

Comment 3:In the Results section, there is a confusing sentence: The average age range of participants is 41 to 50 years. The average or mean is a value, while the range is an interval. 

Response 3:We rephrased this sentence. This can be found in Lines 97 & 98.

Comment 4: Gathering data in a cold season psychologically encourages participants to recall recent impressions and less the experience at temperatures of more than 30 degrees C. In my opinion, it could be a bias. 

Response 4: We added this in the limitations section. This can now be found in lines 209 – 211.

Comment 5: The discussion section is briefly covered, with the most cited article of your own. Perhaps it can be explained by the novelty of your approach or the scarcity of literature on green exercise...

Response 5: We incorporated additional literature from other fields into the discussion to enhance this section. Since YRGE has not been researched extensively

Comment 6: Including the Abbreviation section at the end of the article will be helpful for readers.

Response 6: This can now be found in Lines 212 - 219

We hope that we have understood your comments correctly and made the necessary changes accordingly.

Thank you very much for your support and your contributions to our research.

Kind regards

The Authors

Reviewer 2 Report

Comments and Suggestions for Authors

Dear researchers,

Congratulations on your research. Here are my suggestions and comments on the paper:

1.- Introduction

Lines 31-31. What are these benefits? Are they transferable to areas of daily life? Are these benefits acquired by the entire population or only by a particular group?

What happens with the atmospheric climate (temperature) and GE?

2.- Methods

I understand that there have been previous studies on this topic. However, it is necessary to establish the sub-items that make up this section, including design, sample, procedures, instruments, and ethical considerations, among others.

Lines 57–58. I think it is important to point out (as I asked above) how PA practice can vary depending on weather conditions. As well as the variability of ambient temperature.

Lines 59–65. The instrument is mentioned. How was the instrument constructed? Was it adapted? Etc.

3.- Results

For a better appreciation of the results, I believe it is necessary to present them in tables. I believe that beyond not having clear selection criteria (only that they practice GE), everyone can participate. Another important point is to offer a practical definition of GE. They have pointed out that it is “similar to physical activity in nature,” but we know that there is a big difference between PA and exercise.

An objective and at least three questions have been presented in the introduction. These should be key to the presentation of the results, the development of the discussion, and the conclusion.

5.- Conclusions

They have presented an objective and at least three questions in the introduction (How are they related to each other?

6.- References

I tried to review some articles, and it led me to other articles by the authors. Because they have not provided DOIs or links

Author Response

Dear Reviewer 2

We thank you for your constructive and helpful comments on the manuscript. We have tried to implement them to your satisfaction.

The changes that we conducted are highlighted in yellow within the manuscript

1.- Introduction

Comment 1: Lines 31-31. What are these benefits? Are they transferable to areas of daily life? Are these benefits acquired by the entire population or only by a particular group?

Response 1:We added information about these benefits. These changes can now be found in lines 31 & 32 as well as in lines 35 & 36

Comment  2:What happens with the atmospheric climate (temperature) and GE?

Response 2: Declining temperature leads to a reduction in GE. We added this information, which can be found in line 42

2.- Methods

Comment 3: I understand that there have been previous studies on this topic. However, it is necessary to establish the sub-items that make up this section, including design, sample, procedures, instruments, and ethical considerations, among others.

Response 3: In the Materials and Methods section, we provide an overview of how the study was conducted, along with a brief description of the participants and a link to the study where detailed information can be found.

Comment 4: Lines 57–58. I think it is important to point out (as I asked above) how PA practice can vary depending on weather conditions. As well as the variability of ambient temperature.

Response 4: We added information. This can now be found in lines 44 & 45.

Comment 5: Lines 59–65. The instrument is mentioned. How was the instrument constructed? Was it adapted? Etc.

Response 5: We added information on how the instrument was conducted. This change can now be found in line 66

3.- Results

Comment 6. For a better appreciation of the results, I believe it is necessary to present them in tables. I believe that beyond not having clear selection criteria (only that they practice GE), everyone can participate. Another important point is to offer a practical definition of GE. They have pointed out that it is “similar to physical activity in nature,” but we know that there is a big difference between PA and exercise.

Response 6: We added a table, which can now be found in line 149. We also added some information to clarify the understanding of GE, which was used, which can be found in lines 35 &78

Comment 7: An objective and at least three questions have been presented in the introduction. These should be key to the presentation of the results, the development of the discussion, and the conclusion.

Response 7: We added an explanation of why we chose these questions and how they are related to each other to the beginning of the discussion

5.- Conclusions

Comment 8: They have presented an objective and at least three questions in the introduction (How are they related to each other?

Response 8: We added information to help readers understand how the research questions are related to each other. This can be found in lines 155 - 158

6.- References

Comment 9: I tried to review some articles, and it led me to other articles by the authors. Because they have not provided DOIs or links

Response 9: Which articles were you not able to find? If you could provide that information, we would gladly check if we made some mistakes with the citation of these articles

We hope that we understood your comments correctly and changed them accordingly.

Thank you very much for your support and your contributions to our research.

Kind regards

The Authors